# Numerical Investigation of Fluid Flow and Heat Transfer in High-Temperature Wavy Microchannels with Different Shaped Fins Cooled by Liquid Metal

**DOI:** 10.3390/mi14071366

**Published:** 2023-07-02

**Authors:** Tingfang Yu, Xing Guo, Yicun Tang, Xuan Zhang, Lizhi Wang, Tao Wu

**Affiliations:** 1Department of Energy and Power Engineering, School of Advanced Manufacturing, Nanchang University, Nanchang 330031, China; yutingfang@ncu.edu.cn (T.Y.); xingg@email.ncu.edu.cn (X.G.); 2Department of Energy and Power Engineering, School of Mechanical Engineering, Beijing Institute of Technology, Beijing 100081, China; 3Yangzhou Collaborative Innovation Research Institute Co., Ltd., Yangzhou 225006, China; lychee@mail.ustc.edu.cn (L.W.); wutao_921006@163.com (T.W.)

**Keywords:** microchannel, heat transfer augmentation, wavy channel, fin shape, liquid metal

## Abstract

The microchannel heat sink has been recognized as an excellent solution in high-density heat flux devices for its high efficiency in heat removal with limited spaces; however, the most effective structure of microchannels for heat dissipation is still unknown. In this study, the fluid flow and heat transfer in high-temperature wavy microchannels with various shaped fins, including the bare wavy channel, and the wavy channel with circular, square, and diamond-shaped fins, are numerically investigated. The liquid metal-cooled characteristics of the proposed microchannels are compared with that of the smooth straight channel, with respect to the pressure drop, average Nusselt number, and overall performance factor. The results indicate that the wavy structure and fin shape have a significant effect on the heat sink performance. Heat transfer augmentation is observed in the wavy channels, especially coupled with different shaped fins; however, a large penalty of pressure drops is also found in these channels. The diamond-shaped fins yield the best heat transfer augmentation but the worst pumping performance, followed by the square-, and circular-shaped fins. When the *Re* number increases from 117 to 410, the *Nu* number increases by 61.7% for the diamond fins, while the ∆*p* increases as much as 7.5 times.

## 1. Introduction

The microchannel heat sink is encountered in many industrial applications such as chip cooling, aerospace engineering, micro-electromechanical systems, and microfluidic control devices [1,2,3,4]. Due to the advantage of a high heat transfer coefficient and an enlarged heat transfer area, the microchannel heat sink is recognized as an excellent heat removal device with limited spaces. Tuckerman and Peace [5] introduced the first microchannel heat sink, which was fabricated on a silicon wafer and used water as the liquid coolant, where the heat flux that was dissipated was 790 W/cm^2^. Since then, much attention has been focused on the improvement of the cooling performance for the microchannel heat sink [6,7,8,9,10,11]. Steinke and Kandlikar [12] made a systematic review of the single-phase flow friction factors in microchannels, and they compared the experimental data to discover the deviations in the literature. Smakulski and Pietrowicz [13] compared the cooling techniques in microchannel heat sinks, by considering the coolant type, pressure drop, maximal heat flux, and heat transfer coefficient. They found that spray cooling could enhance the heat transfer significantly more than normal, single-phase flow cooling.

The microchannel configurations have a direct effect on the performance of the heat sink, and many novel designed microchannels have been proposed in the literature, which includes secondary branches, different shaped cross-sectional shapes, various inlets and outlets, different flow passage types, and porous fillings. Xia et al. [14] investigated the effects of different inlet and outlet positions, and channel shapes on the performance of microchannel heat sinks, and they found that the I-type flow passage with a rectangular-shaped channel achieved the optimal performance. Gong et al. [15] numerically investigated the effects of metallic, porous fins on the microchannel heat sink performance, and they concluded that the porous fins could deteriorate the heat transfer because the bulk flow velocity decreased, but that the pressure drop was lowered because shear stress at the solid–liquid interface was reduced.

The coolant type also influences the microchannel heat sink performance greatly due to excellent thermal properties. Some commonly used coolants, such as water, ethyl alcohol, and ethylene glycol, cannot keep up with the growing demand of the cooling capacity for high-density heat flux devices; therefore, many researchers have paid attention to new kinds of coolants, such as nanofluids and liquid metals [16,17,18,19,20,21]. Nanofluids can achieve a better heat performance because of their enlarged heat transfer area brought about by the numerous nanoparticles. Salman et al. [17] investigated the influences of nanoparticle and base fluid types on the heat dissipation capacity of microchannel heat sinks, and they found that SiO_2_ nanoparticles suspended in ethylene glycol achieved the largest Nusselt number, followed by ZnO, CuO, and Al_2_O_3_. Ho et al. [18] investigated the heat transfer of a microchannel heat sink with nanofluids experimentally, and concluded that the thermal resistance was cut down by 12.61%, and the convective heat transfer coefficient increased by 14.43%.

Liquid metal is considered the most appropriate coolant for high-temperature cooling situations for its high thermal conductivity, electrical conductivity, and mobility [22]. Wu et al. [23] numerically investigated a silicon carbide-fabricated heat sink with different shaped cross-sectional microchannels, and cooled by different liquid metal fluids, including sodium, potassium, a sodium–potassium alloy, and lithium. They found that the liquid lithium with circular-shaped microchannels had the best heat dissipation performance among all the microchannels studied. Ghoshal et al. [24] investigated a GaIn alloy loop-cooled heat sink, and found that the mean heat transfer coefficient could achieve 20 W/cm^2^∙K, with the thermal resistance lowered to 0.22 K/W. Tawk et al. [25] studied a rectangular-shaped channel cooler with a gallium alloy as the coolant, and concluded that the liquid metal could efficiently enhance the heat removal for high-density heat flux electronic devices.

From the literature review, it can be concluded that novel configuration designs of microchannel heat sinks can achieve better heat dissipation performances. In addition, the proper coolant type has a significant influence on the heat transfer and fluid flow characteristics in the microchannels; however, the most efficient structure of microchannels for heat dissipation is still unknown. In this paper, the laminar fluid flow and heat transfer in high-temperature wavy microchannels with various shaped fins are numerically investigated, including the bare wavy channel, and the wavy channel with circular-, square-, and diamond-shaped fins, while liquid lithium is chosen as the coolant. Special attention is paid to the comparison between the proposed microchannel heat sinks with the smooth straight channel, according to the pressure drop, average friction coefficient, average Nusselt number, and the overall performance factor. The effects of the fin relative size and fin relative height on the fluid flow and heat transfer characteristics of the microchannel heat sink are also investigated.

## 2. Theoretical Model and Numerical Procedure

### 2.1. Physical Model

The 3D-microchannel heat sink studied in this work is illustrated schematically in Figure 1a, whose length is *L*, width is *W*, and height is *H*. The smooth, straight channel heat sink has six parallel microchannels with rectangular cross-sectional shapes. The microchannel has a length of *L*, width of *W*_*c*_, and height of *H*_*c*_, while the thickness between the adjacent channels is *W*_*w*_. The microchannels are fabricated on a silicon carbide material, and the base thickness is *H*_*w*_. The coordinate system is also shown in Figure 1a. The coolant fluid flows toward the positive *x*-direction with a uniform inflow velocity. At the bottom surface, the microchannel heat sink is heated by a uniform and constant heat flux. Figure 1b shows the four new microchannels proposed in this study, including the wavy channel without fins, and the wavy channel with circular-, square-, and diamond-shaped fins. The wavy channel is sinusoidal-shaped with a wavelength of *L*_*t*_ and amplitude of *W*_*t*_. The different shaped fins have the same hydraulic diameter *D*_*h*_ and fin height *H*_*b*_. The geometric parameters are listed in Table 1. The computational domain of the physical model and the relevant boundary conditions are depicted in Figure 1c, which only includes one microchannel due to the symmetric feature and the consideration of saving computation resources.

### 2.2. Mathematical Model

The following assumptions are made in order to develop the mathematical model:(1)The flow is laminar and steady state, and viscous dissipation is neglected;(2)The radiative heat transfer, surface tension, and gravitational force are not considered;(3)The density, dynamic viscosity, specific heat capacity, and thermal conductivity of the fluid are functions of the temperature, and the thermal properties of the solid are supposed to be constant.

The laminar assumptions for the fluid flow in wavy channels are indicated by Sui et al. [26], in which the wavy microchannel with a rectangular-shaped cross-section was investigated, and they found that the flow state was laminar in wavy channels when *Re* < 800. In the present study, the *Re* number ranges from 117 to 410; therefore, the laminar assumption can be accepted in this regard.

Based on the assumptions above, the mass conservation equation can be written as:(1)$$\frac{\partial }{\partial x_{i}}\left(\rho_{f}u_{i}\right)=0,\;i,j=1,2,3$$

The momentum equation can be expressed by:(2)$$\frac{\partial }{\partial x_{i}}\left(\rho_{f}u_{i}u_{j}\right)=-\frac{\partial p}{\partial x_{j}}+\frac{\partial }{\partial x_{i}}\left[\mu_{f}\left(\frac{\partial u_{j}}{\partial x_{i}}+\frac{\partial u_{i}}{\partial x_{j}}\right)\right],\;i,j=1,2,3$$

The energy equation for the fluid can be given by:(3)$$\frac{\partial }{\partial x_{i}}\left(\rho_{f}u_{i}c_{p,f}T_{f}\right)=\frac{\partial }{\partial x_{i}}\left(\lambda_{f}\frac{\partial T_{f}}{\partial x_{i}}\right),\;i,j=1,2,3$$

The energy equation for the solid is as follows:(4)$$\frac{\partial }{\partial x_{i}}\left(\lambda_{s}\frac{\partial T_{s}}{\partial x_{i}}\right)=0,\;i,j=1,2,3$$
where *i*, *j* = 1, 2, 3 represent the *x*, *y*, and *z* directions, respectively. *u_i_* (*i* = 1, 2, 3) stands for the velocity component in the *x*, *y*, and *z* direction, respectively. *T* depicts the temperature, and *p* is the static pressure. *ρ* stands for the fluid density, *μ* is the dynamic viscosity, and *c_p_* and *λ* are the specific heat capacity and thermal conductivity, respectively. The subscripts *f* and *s* denote the fluid and solid, respectively. The thermal properties of the silicon carbide is specified as follows: *ρ_s_* = 3220 kg/m^3^, *c*_*p*,*s*_ = 800 J/(kg∙K), and *λ_s_* = 80 W/(m∙K). The thermal properties of the liquid lithium are defined as follows [23]:(5)$$\rho_{f}=278.5-0.04657T+274.6{\left(1-T/3500\right)}^{0.467}$$
(6)$$c_{p,f}=4754-0.925T+2.91\times 10^{-4}T^{2}$$
(7)$$\lambda_{f}=22.28+0.05T-1.243\times 10^{-5}T^{2}$$
(8)$$\mu_{f}=\exp\left[-4.16435-0.6374\ln\left(T\right)+292.1/T\right]$$

Boundary conditions:(1)At the channel inlet (*x* = 0), the inlet velocity and fluid temperature are specified as:(9)$$u=u_{\mathrm{in}},\;v=0,\;w=0;\;T_{f}=T_{\mathrm{in}}$$
where *u*_in_ is set to be 0.1, 0.15, 0.2, 0.25, 0.3, and 0.35 m/s, and the Reynolds number is 117.1, 175.7, 234.3, 292.8, 351.4, and 410.0, respectively. The inlet temperature is *T*_in_ = 600 K.(2)At the channel outlet (x = L), the pressure is defined as the atmospheric pressure:(10)$$p_{\mathrm{out}}=p_{\mathrm{atm}}$$
where *p*_atm_ uses the gauge pressure at the channel outlet, and *p*_atm_ = 0 Pa.(3)At the solid–fluid interface, non-slip and coupled boundaries are applied:(11)$$u=0,v=0,w=0;\;T_{f}=T_{s};\;-\lambda_{s}\frac{\partial T_{s}}{\partial n}=-\lambda_{f}\frac{\partial T_{f}}{\partial n}$$
where *n* denotes the normal direction.(4)At the bottom surface (*z* = 0), a uniform and constant heat flux $q^{\prime \prime }$ is applied:(12)$$-\lambda_{s}\frac{\partial T_{s}}{\partial z}=q^{\prime \prime }$$
where $q^{\prime \prime }$ is set to be 200 W/cm^2^ in this study.(5)At the symmetric planes (*y* = 0, and *y* = *W_c_* + 2*W_w_*), symmetric boundary conditions are used:(13)$$-\lambda_{s}\frac{\partial T_{s}}{\partial y}=0$$(6)For other surfaces, the adiabatic boundaries are used:(14)$$-\lambda_{s}\frac{\partial T_{s}}{\partial n}=0;\;-\lambda_{f}\frac{\partial T_{f}}{\partial n}=0$$

### 2.3. Numerical Work

The meshing procedure for the computational domain was performed in the commercial software ICEM CFD 19.0, which utilizes unstructured hexahedral cells to discretize the whole model. Then, the mesh was imported to the software ANSYS Fluent 19.0 for further setting. The coupled pressure-velocity field was decoupled using the SIMPLEC algorithm, which is more stable and can obtain a faster convergence. The convection and diffusion terms in the PDEs were discretized with the second order upwind and central difference scheme, respectively. The convergence criterions for the related parameters were specified as 10^−5^, except for the energy equation, which was specified as 10^−7^.

Grid independence testing was performed to obtain mesh-free results. The pressure drop and average bottom temperature were used as the parameters in this study, with the inlet velocity set as 0.25 m/s. Four grids were generated for the smooth straight channel, 0.54, 0.87, and 1.22 million grids were compared with 2.58 million grids, and the pressure drop deviated by 6.99%, 6.15%, and 0.35%, respectively. In addition, the average bottom temperature deviated by 0.019%, 0.018%, and 0.001%, respectively; therefore, 1.22 million grids were chosen for calculation in the study. Furthermore, 0.75, 1.03, and 1.67 million grids were compared with 2.57 million grids for the wavy channel with diamond fins, and the pressure drop deviated by 8.36%, 4.41%, and 2.07%, while the average bottom temperature deviated by 0.034%, 0.022%, and 0.007%, respectively; thus, 1.67 million grids were used for further study. Using the same method, the final grids for the bare wavy channel, and the wavy channel with circular-, and square-shaped fins were 1.37, 1.63, and 1.64 million, respectively.

### 2.4. Data Reduction

The Reynolds number is a basic parameter for evaluating the flow regime, which is calculated from:(15)$$Re=\frac{\rho_{f}u_{in}D_{h}}{\mu_{f}}$$
where *u*_in_ is the inlet flow velocity. *ρ_f_*, and *μ_f_* stand for the density and dynamic viscosity of the fluid, respectively. *D_h_* depicts the hydraulic diameter, which is defined as:(16)$$D_{h}=\frac{2H_{c}W_{c}}{H_{c}+W_{c}}$$

The mean convective heat transfer coefficient is given by:(17)$$h=\frac{q^{\prime \prime }A_{b}}{A_{\mathrm{con}}\left(T_{w}-T_{f}\right)}$$
where *T_w_* is the average temperature of the heating surface, and *T_f_* stands for the average temperature of the fluid. *A_b_* and *A*_con_ are the areas of the heating surface, and the contacting surface between the solid and fluid, respectively.

The average Nusselt number and average friction coefficient are calculated from:(18)$$Nu=\frac{hD_{h}}{\lambda_{f}}$$
(19)$$f=\frac{D_{h}\Delta p}{2L\rho_{f}u_{in}^{2}}$$
where *λ_f_* represents the fluid thermal conductivity, and Δ*p* is the static pressure difference between the microchannel inlet and outlet.

The overall performance factor is calculated from [27]:(20)$$\eta =\frac{Nu/Nu_{0}}{{\left(f/f_{0}\right)}^{1/3}}$$
where *Nu*_0_ and *f*_0_ stand for the heat transfer and fluid flow parameters of the smooth straight channel, respectively. 

The field synergy number is an effective parameter to distinguish the synergistic effect between the velocity and temperature field. In this paper, the field synergy number is adopted to evaluate the heat transfer enhancement in the newly-proposed microchannels, which is given by [28]:(21)$$F_{c}={∭}_{V_{c}}\left(U_{\mathrm{ave}}\cdot \nabla T_{f,\mathrm{ave}}\right)dV_{\mathrm{ave}}=\frac{Nu}{RePr}$$
where *U*_ave_ stands for the average velocity vector, *T*_*f*,ave_ is the average fluid temperature, and *V*_ave_ depicts the fluid volume. *V_c_* is the control volume. *Pr* stands for the Prandtl number, which is defined as *Pr* = *μ_f_*∙*c*_*p*,*f*_/*λ_f_*.

Bejan [29] used the concept of the volume entropy generation rate to evaluate the irreversible heat loss for the energy quality occurring in the microchannel heat sinks, which consists of two kinds of entropy generation in the fluid flow and heat transfer process, i.e., the friction entropy generation rate and the heat transfer entropy generation rate. The definitions of these two parameters are calculated from:(22)$$S_{\mathrm{gen},\Delta p}=∭{S^{\prime \prime \prime }}_{\mathrm{gen},\Delta p}dV=\frac{m}{\rho_{f}T_{f}}\Delta p$$
(23)$$S_{\mathrm{gen},\Delta T}=∭{S^{\prime \prime \prime }}_{\mathrm{gen},\Delta T}dV=\frac{q^{\prime \prime }A_{\mathrm{con}}}{T_{\mathrm{con}}T_{f}}\left(T_{\mathrm{con}}-T_{f}\right)$$

### 2.5. Model Validation

In order to validate the reliability of the numerical model, the present numerical results were compared with the experimental data in the literature. Figure 2 illustrates the comparisons between the present numerical results and the experimental data from Chai et al. [30], in terms of the average Nusselt number and the product of average friction coefficient and Reynolds number. It was found that the present numerical results were in reasonable agreement with the experimental data, where the maximum deviation was 12.77% for the average Nu number, and 3.86% for the product of the average friction coefficient and *Re* number. The relative discrepancy for the average *Nu* number might have resulted from the radiative and convective heat transfer between the heat sink and the surroundings. The comparison implies that the numerical procedure can further be performed to investigate the fluid flow and heat transfer in the newly-proposed microchannel heat sinks.

## 3. Results and Discussion

### 3.1. Fluid Flow Characteristics

To observe the velocity, streamline, pressure, and temperature distribution in the microchannels, a horizontal plane parallel to the bottom surface with a distance *z* = 2.5 mm was extracted. Figure 3 shows the velocity and streamline distributions at *z* = 2.5 mm in the microchannels for *Re* = 292.8. The plots are zoomed at the *x* locations in the range of 7.5 < *x* < 9.5 mm. Figure 3 indicates that the velocity distribution presented a parabolic type flow in the smooth straight channel, with the streamlines in parallel with each other; however, the velocity distributions in the four new microchannels showed obvious differences with that of the smooth straight channel. For the wavy microchannel without fins, the fluid in the middle of the microchannel had a higher velocity, while the fluid near the sidewalls had a lower velocity. The velocity contour presented a periodic and wavy distribution, and the streamlines were parallel with each other in the middle of the channel, but periodic vortexes were found in the cavities on the sidewalls. The vortex sizes in the cavities were almost the same because of the equal-sized cross-sections in the wavy microchannel.

For the wavy microchannel with different shaped fins, the velocity and streamline distributions were more complex than the bare wavy channel. The fins in the middle of the channel brought about a strong blocking effect to the fluid flow, which separated the fluid into two parts, where one part flowed into the left side of the fins and the other flowed into the right side. Due to the asymmetry of the cross-sectional areas at the two sides of the fins, the separated fluids flowed around the fin with different velocities; thus, additional vortexes were formed behind the fins. The size of the additional vortexes depended on the fin shapes, where the diamond-shaped fin brought about the largest additional vortexes, followed by the square-shaped fin. The circular-shaped fin brought almost no additional vortexes because of its streamlined structures. The vortexes were also found in the cavities on the sidewalls for the wavy channel with different shaped fins. The vortex size was larger at the location where the fluid velocity was smaller. The different shaped fins could bring about a strong flow disturbance and fluid mixing; thus, the flow boundary layers in the channels were interrupted and redeveloped periodically along the flow direction.

Figure 4 illustrates the pressure distribution at *z* = 2.5 mm in the microchannels for *Re* = 292.8. It was found that the pressure in the smooth straight channel was obviously smaller than those in the wavy channels. The wavy channel with fins showed a slightly higher pressure drop than the smooth straight channel due to the periodic cavities on the sidewalls. For the wavy channel with different shaped fins, the pressure drop increased rapidly because of the blocking effect induced by the fins. The diamond-shaped fin brought about the highest pressure drop, followed by the square- and circular-shaped fins. The circular-shaped fins had a minimal blocking effect on the fluid because of their streamlined structures; however, the diamond-shaped fins greatly changed the direction of the main stream fluid, which had the maximum blocking effect on the fluid.

To have a quantitative comprehension of the blocking effects caused by the different fins, the changes in the Δ*p* with the *Re* number is depicted in Figure 5a. The results show that the pressure drops increased with the *Re* number for all the microchannels in this study. In the range of 117.1 < *Re* < 410.0, the pressure drops increased from 35.3 Pa to 157.3 Pa for the smooth straight channel, from 71.8 Pa to 409.2 Pa for the wavy channel without fins, from 318.7 Pa to 2089.9 Pa for the wavy channel with circular-shaped fins, from 545.2 Pa to 3909.1 Pa for the square-shaped fins, and from 592.6 Pa to 5040.9 Pa for the diamond-shaped fins. The pressure drop for the diamond-shaped fin increased drastically with the *Re* number, which was far more than the other microchannels.

The changes in the average friction coefficient with the Reynolds number are demonstrated in Figure 5b. The average friction coefficient was calculated from Equation (19). It can be found that the average friction coefficient declined with the Re number for all the cases. This can be explained by Equation (19), in which the average friction coefficient is strongly affected by the inlet velocity with a quadratic relation; thus, the average friction coefficient is smaller at high *Re* numbers. In the range of 117.1 < *Re* < 410.0, the average friction coefficient declined from 0.178 to 0.064 for the smooth straight channel, from 0.361 to 0.167 for the wavy channel without fins, from 1.601 to 0.852 for the wavy channel with circular-shaped fins, from 2.739 to 1.593 for the square-shaped fins, and from 2.978 to 2.054 for the diamond-shaped fins.

### 3.2. Heat Transfer Characteristics

The temperature distribution at *z* = 2.5 mm in the microchannels for *Re* = 292.8 is shown in Figure 6. It can be found that the fluid temperature increased along the flow direction for all the microchannels. The figure clearly depicts the temperature difference at the solid–fluid interface, which reveals the temperature gradient and the thermal conductivity difference between the solid and the fluid phase. The temperature for the wavy channel with different shaped fins showed as slightly lower than the smooth straight channel, and the reason for this can be explained by the enlarged heat transfer area for the wavy microchannels and different fins. Moreover, the fins could have brought about periodic interruptions for the thermal boundary layers in the microchannels, and the local heat transfer was enhanced.

The variations of the average Nusselt number is illustrated in Figure 7a. It can be found that the average *Nu* number increased with the *Re* number for all the cases. When the *Re* number increased from 117.1 to 410.0, the average *Nu* number increased from 20.09 to 27.30 for the smooth straight channel, from 22.79 to 28.29 for the wavy channel without fins, from 23.41 to 35.76 for the wavy channel with circular-shaped fins, from 23.51 to 38.29 for the square-shaped fins, and from 23.97 to 38.76 for the diamond-shaped fins. The results indicate that the wavy channel with different fins had a significant impact on the heat transfer. The diamond-shaped fin achieved the highest heat transfer augmentation among all the microchannels studied in this work.

Analyses were further carried out to investigate the overall performance factor for the different newly-proposed microchannels, which considered not only the heat transfer augmentation but also the pressure drop penalty. The overall performance factors under different *Re* numbers are given in Figure 7b, in which the overall performance factors were compared with the smooth straight channel. It can be found that the wavy channel with different fins achieved a more deteriorative overall performance factor than that of the wavy channel without fins, and the reason for this is that the blocking effects brought about by the different shaped fins were more severe than the heat transfer augmentation. Interestingly, the overall performance factor for the wavy channel with different shaped fins reached a peak at *Re* 240, which was induced by the relative dominance between the friction drag and the heat transfer augmentation. The heat transfer augmentation dominated at lower *Re* numbers but the friction drag greatly dominated at higher *Re* numbers. The overall performance factor ranged from 44.7% to 47.7% for the wavy channel with diamond-shaped fins, which was the lowest among all the microchannels.

The field synergy number *Fc* under different *Re* numbers is demonstrated in Figure 7c. It is plain to see that the field synergy number declined with the *Re* numbers for all the microchannels. The field synergy number declined from 5.0 to 1.8 for the smooth straight channel, from 5.7 to 1.8 for the wavy channel without fins, from 5.8 to 2.3 for the wavy channel with circular-shaped fins, from 5.8 to 2.5 for the square-shaped fins, and from 5.9 to 2.5 for the diamond-shaped fins. Among all the microchannels studied in this work, the diamond-shaped fin brought about the highest *Fc*. The reason for this can be explained by the synergy angle between the velocity vector and the temperature gradient. The diamond-shaped fin possessed the smallest angle due to the shape structure among all the microchannels; thus it achieved the highest field synergy number.

The changes in the friction and heat transfer entropy generation rate with the *Re* number are shown in Figure 8. In Figure 8a, the friction entropy generation rate increases with the *Re* number for all the microchannels. The wavy channel with diamond-shaped fins achieved the largest friction entropy generation rate, then followed by the square- and circular-shaped fins. This can be explained by the strong blocking effects brought about by the fins. In Figure 8b, the heat transfer entropy generation rate for the wavy channel with different shaped fins decreases with the *Re* number, and the reason for this can be explained by the reduced temperature difference between the fluid and solid when the *Re* number was larger. The heat transfer entropy generation rate, however, for the smooth straight channel and wavy channel without fins increased with the *Re* number at first, then decreased at higher *Re* numbers, and the reason for this can also be attributed to the temperature difference variation between the solid and fluid, which was increased at first due to the shortened retention time for the heat transfer, but was decreased for the strong fluid mixing at higher *Re* numbers in the microchannels.

### 3.3. Effect of the Fin Relative Size

The fin relative size plays an important role in the heat transfer and fluid flow characteristics due to the constricted cross-sectional areas brought about by the fins. It is interesting to investigate the influence of the fin relative size on the comprehensive performance of the microchannel heat sinks. In this work, the fin height was maintained at 3 mm, while the fin relative size ratio was set to be 0.2, 0.4, 0.6, and 0.8, respectively. The wavy channel with square-shaped fins was chosen as the calculated microchannel, and the fin sizes were 0.04, 0.08, 0.12, and 0.16 mm, respectively. They were compared with the original microchannel, with the fin relative size set as 1.0, and the fin size as 0.2 mm.

Figure 9 demonstrates the changes in the average friction coefficient, average *Nu* number, and the overall performance factor with the fin relative size. Figure 9a shows that the average friction coefficient increased with the fin relative size, which can be explained by the constricted flow area and the increasing flow velocity and blocking effect brought about by the fins. It can also be noted that the average friction coefficient decreased with the *Re* number for a certain fin size, which can be explained by Equation (19), where the friction coefficient declined with the fluid velocity.

In Figure 9b, the average *Nu* number tends to increase with the fin relative size, and the average *Nu* number is larger under a larger *Re* number. This is reasonable because the average heat transfer coefficient was higher when the fluid flowed at a higher speed, and the heat dissipation was more reinforced. In Figure 9c, the overall performance factor decreased with fin relative size, because the average friction coefficient increased more rapidly than the heat transfer enhancement, leading to a declined overall performance factor.

### 3.4. Effect of the Fin Relative Height

The fin relative height also has a significant impact on the heat transfer and fluid flow of the microchannel heat sinks due to an enlarged heat transfer area, along with the accompanying blocking effects by the fins. In this work, the fin size was kept at 0.2 mm, and the fin relative height ratio was set to be 0.2, 0.4, 0.6, and 0.8, respectively. The wavy channel with square-shaped fins was chosen as the calculated microchannel, and the fin heights were 0.6, 1.2, 1.8, and 2.4 mm, respectively. They were compared with the original microchannel, with the fin relative size set as 1.0, and the fin size as 3.0 mm.

Figure 10 depicts the variations in the average friction coefficient, average *Nu* number, and the overall performance factor with the fin relative height. In Figure 10a, it obviously shows that the average friction coefficient increased with the fin relative height, and the reason for is that the blocking effect was more intensive with a higher fin height. In Figure 10b, it is interesting to note that the average *Nu* number decreases with the fin relative height at first, then increases rapidly with the fin height. This can be explained by the relative dominance between the friction drag and the enlarged heat transfer area. When the fin relative height ratio was lower than 0.4, the friction drag dominated while the heat transfer was deteriorated with a larger fin height; however, the heat transfer augmentation dominated when the fin relative height ratio exceeded 0.4, and the average *Nu* number increased with the fin relative height ratio due to the enlarged heat transfer area. In Figure 10c, the overall performance factor decreased with the fin relative height at first, then rose quickly with the fin relative height, which was inconsistent with the average *Nu* number. This implies that the fin height should be paid special attention to when designing new microchannel heat sinks.

## 4. Concluding Remarks

In this paper, numerical studies were conducted on the heat transfer and laminar flow in high-temperature wavy microchannels with different shaped fins, including the bare wavy channel, and the wavy channel with circular-, square-, and diamond-shaped fins, while liquid lithium was chosen as the coolant. The pressure drop, average friction coefficient, average Nusselt number, and the overall performance factor of the proposed microchannels for different Reynolds numbers were compared with that of the smooth, straight channel. The main conclusions are summarized as follows:(1)The presence of wavy channels and different shaped fins has an important impact on the heat transfer and fluid flow characteristics. As the *Re* number increases from 117.1 to 410.0, the average Nusselt number increases by 61.7% for the wavy channel with different shaped fins, while the pressure drop increases by 7.5 times.(2)Among the four newly-proposed microchannels, the wavy channel with diamond-shaped fins brings about the best heat transfer enhancement, followed by the square- and circular-shaped fins. The pressure drop brought about by the diamond-shaped fins is far more than the other microchannels in this study.(3)The overall performance factor decreases with the fin relative size for the four investigated microchannels, and the reason for this is that the average friction coefficient increases more rapidly than the heat transfer enhancement. The constricted flow cross-sectional area leads to stronger friction drag effects with larger fin relative sizes.(4)For the four investigated microchannels, the overall performance factor decreases with the fin relative height at first, then increases drastically. The friction drag effect dominates when the fin relative height ratio is less than 0.4, but a heat transfer enhancement prevails when the fin relative height ratio exceeds 0.4, which is induced by the enlarged heat transfer area.(5)The wavy channel with different shaped fins can lead to a strong heat transfer enhancement, but also induces a higher pumping power penalty. These configurations should be paid special attention to where the pumping power should be strictly restricted.

## Figures and Tables

**Figure 1 micromachines-14-01366-f001:**
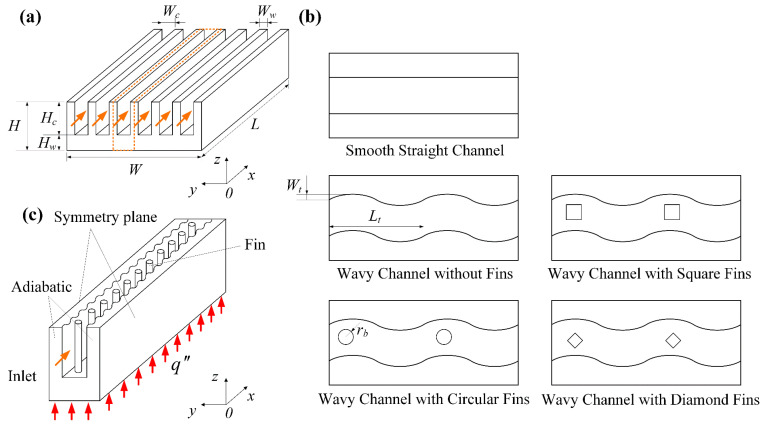
(**a**) Physical model; (**b**) microchannels with different geometric configurations; (**c**) computational domain and boundary conditions.

**Figure 2 micromachines-14-01366-f002:**
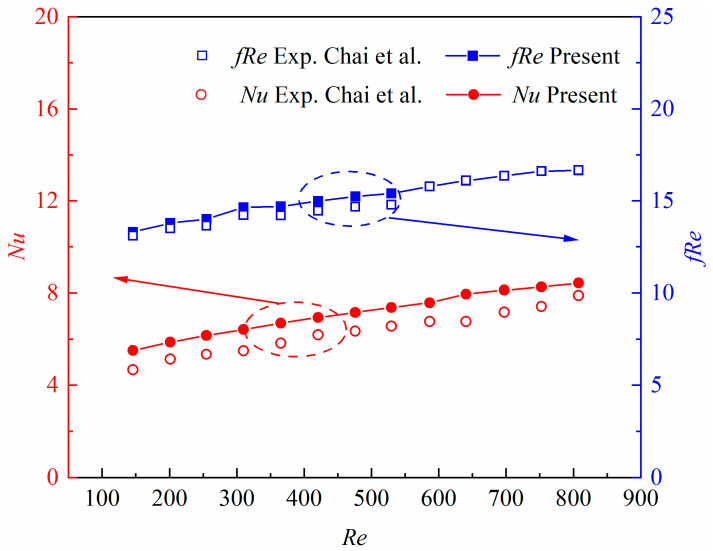
Comparisons of the present numerical results with experimental data from Ref. [30].

**Figure 3 micromachines-14-01366-f003:**
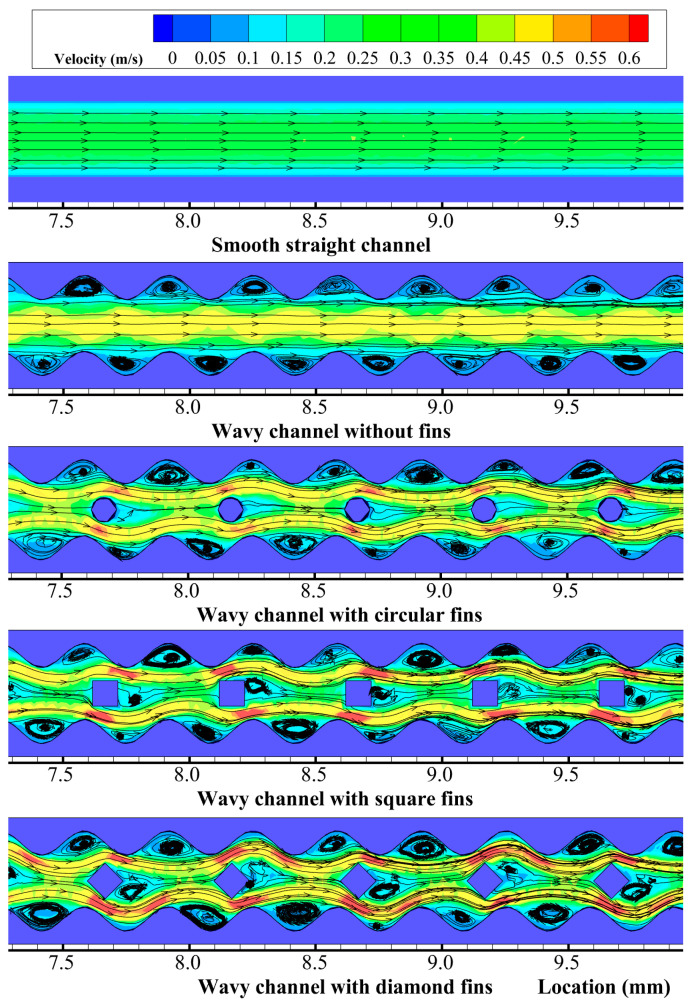
Velocity and streamline distributions at *z* = 2.5 mm in the microchannels for *Re* = 292.8.

**Figure 4 micromachines-14-01366-f004:**
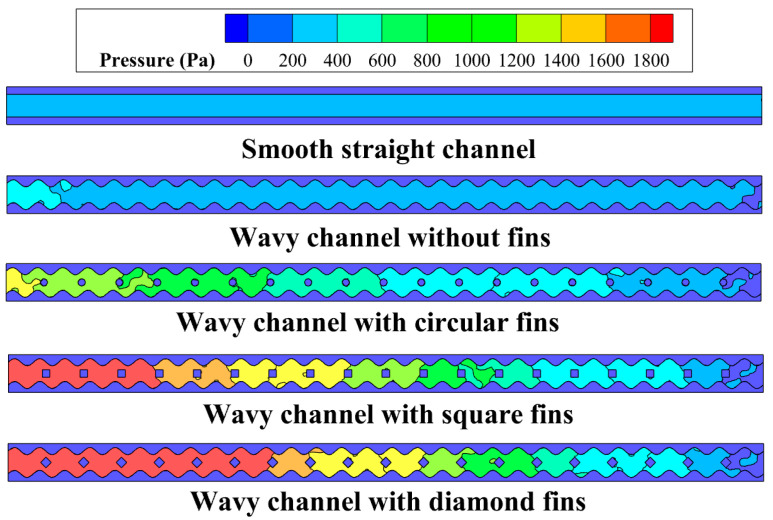
Pressure distribution at *z* = 2.5 mm in the microchannels for *Re* = 292.8.

**Figure 5 micromachines-14-01366-f005:**
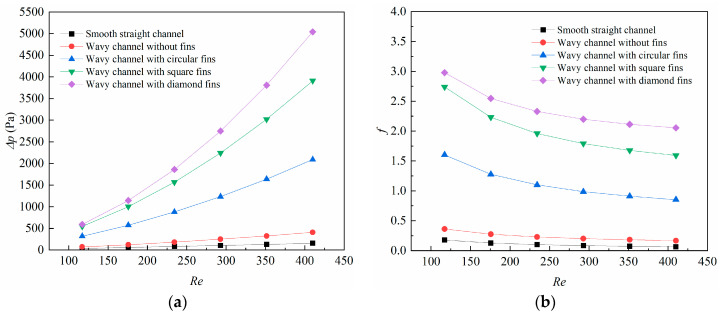
Changes of (**a**) pressure drop (∆*p*); (**b**) average friction coefficient (*f*) with *Re*.

**Figure 6 micromachines-14-01366-f006:**
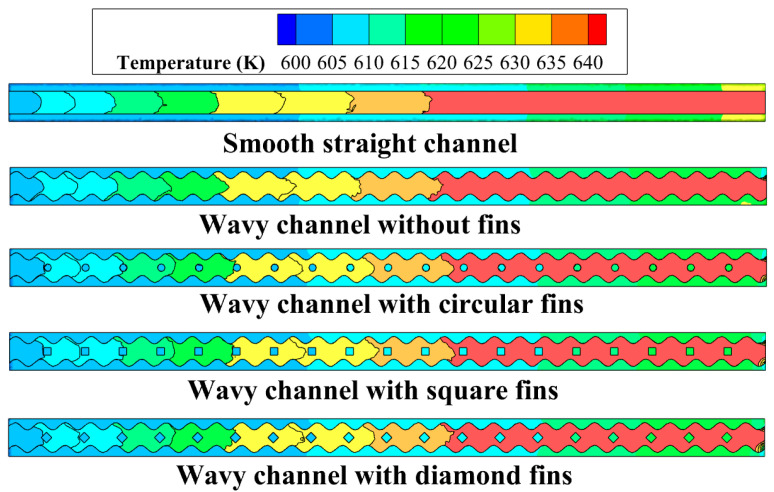
Temperature distributions at *z* = 2.5 mm in the microchannels for *Re* = 292.8.

**Figure 7 micromachines-14-01366-f007:**
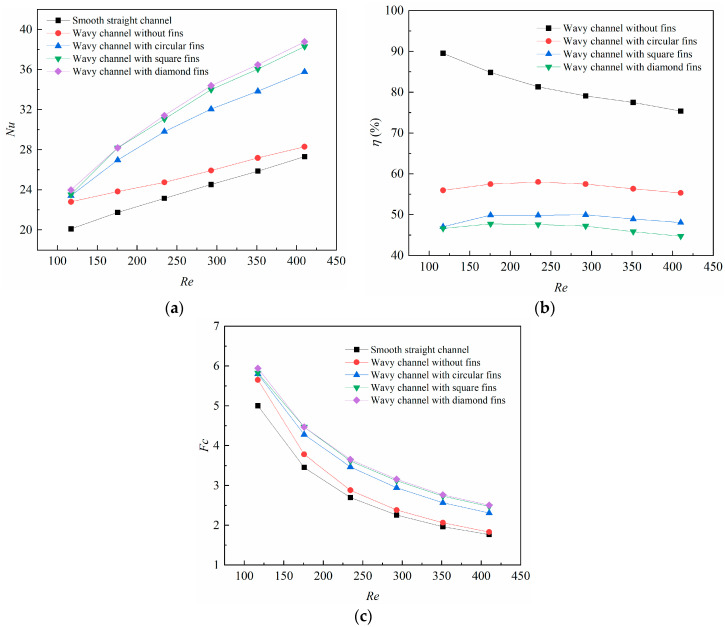
Variations of (**a**) average Nusselt number (*Nu*); (**b**) overall performance factor (*η*); (**c**) field synergy number (*Fc*) with Reynolds numbers.

**Figure 8 micromachines-14-01366-f008:**
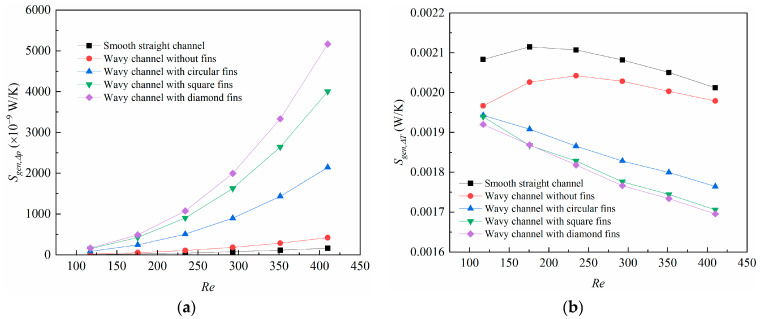
Changes of (**a**) friction entropy generation rate (*S*_gen,Δ*p*_); (**b**) heat transfer entropy generation rate (*S*_gen,Δ*T*_) with Reynolds number.

**Figure 9 micromachines-14-01366-f009:**
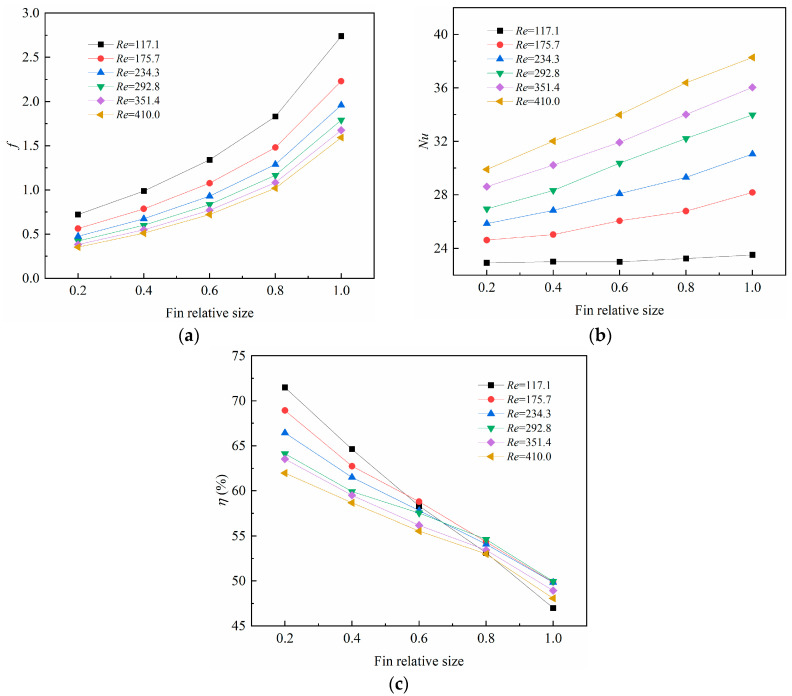
Changes of (**a**) average friction coefficient (*f*); (**b**) average Nusselt number (*Nu*); (**c**) overall performance factor (*η*) with fin relative size.

**Figure 10 micromachines-14-01366-f010:**
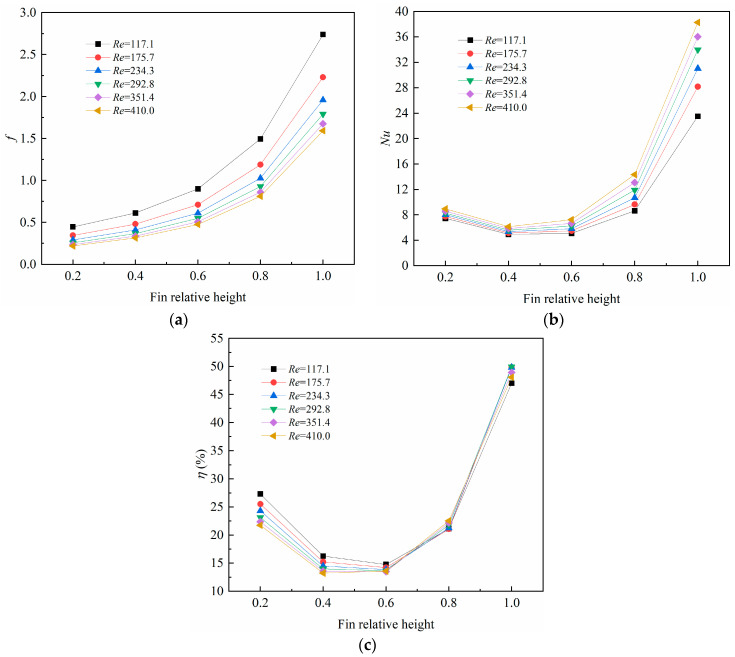
Variations of (**a**) average friction factor (*f*); (**b**) average Nusselt number (*Nu*); (**c**) overall performance factor (*η*) with fin relative height.

**Table 1 micromachines-14-01366-t001:** Geometric sizes used in the present study.

Geometric Parameters	*L*	*W*	*H* _ *c* _	*W* _ *c* _	*W* _ *w* _	*H* _ *w* _	*L* _ *t* _	*W* _ *t* _	*r* _ *b* _	*H* _ *b* _
Value (mm)	20	20	3	0.6	0.4	2	0.3	0.1	0.1	3

## Data Availability

The data presented in this study are available on request from the corresponding author. The data are not publicly available due to privacy.

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
