# Peer review of "Numerical Investigation of Fluid Flow and Heat Transfer in High-Temperature Wavy Microchannels with Different Shaped Fins Cooled by Liquid Metal"

_micromachines, 2023, doi:10.3390/mi14071366_

Round 1

Reviewer 1 Report

The paper studies the fluid flow and heat transfer in high temperature wavy microchannels with various shaped fins investigated. It has some reference value in engineering and I suggest to publish it in this journal after addressing the following problems in the revised paper.

(1)   Generally, turbulent flow is better in heat transfer than laminar flow, why does this paper only pay attention to the latter? Does it have engineering siginicance?

(2)   As a whole, the object studied in this paper is a concrete microchannel. The results obtained may not be general, so the application ranges or a discussion should be given in the conclusions.

Author Response

Response to Reviewer 1 Comments

The paper studies the fluid flow and heat transfer in high temperature wavy microchannels with various shaped fins investigated. It has some reference value in engineering and I suggest to publish it in this journal after addressing the following problems in the revised paper.

Point 1: Generally, turbulent flow is better in heat transfer than laminar flow, why does this paper only pay attention to the latter? Does it have engineering significance?

Response 1: Yes, you are right. Many numerical studies use turbulent models in the simulation, such as Spalatrt-Allmaras, k-epsilon, k-omega model, and so on. Meanwhile, the deionized water is the commonly used coolant fluid in such cooling simulations. The turbulent flow surely can enhance the mean convective heat transfer coefficient, but also brings about high pressure drop penalty, thus lowering the overall performance factor. In our study, liquid metal (liquid lithium) is chosen as the coolant fluid, and the laminar model is utilized in the simulation, which is considered not only from the heat transfer augmentation, but also from the pressure drop penalty. In some applications such as chip cooling and transpiration-cooling nozzle, the coolant velocity is usually lower than other engineering applications for saving the pump power.

Point 2: As a whole, the object studied in this paper is a concrete microchannel. The results obtained may not be general, so the application ranges or a discussion should be given in the conclusions.

Response 2: Yes, the aim of this study is to investigate the effect of the combination of wavy microchannel and different shaped fins on the heat transfer and fluid flow characteristics. Special attentions are paid to the comparison between the proposed microchannel heat sinks with the smooth straight channel. The following statements are added in the revised manuscript:

“…for the four investigated microchannels, …” (Line421)

“For the four investigated microchannels, the…” (Line 425)

Thank you very much for your comments and suggestions!

Prepared by Tingfang Yu, Xing Guo, Yicun Tang, Xuan Zhang, Lizhi Wang, and Tao Wu

June 30, 2023

Reviewer 2 Report

This manuscript indicate that the wavy structure and fin shape have a significance effect on the heat sink performance. Heat transfer augmentation is observed in the wavy channels, especially coupled with different shaped fins. However, large penalty of pressure drops is also found in these channels. The diamond shaped fins yield the best heat transfer augmentation but the worst pumping performance, followed by the square, and circular shaped fins.

Overall, I recommend the manuscript be accepted after minor corrections,

1)pressure drop is an important issue in designing passages of coolant flows to repel high heat flux, but this issue is not discussed clear enough.

2)Explore the possibility and effect of geometric dimensions on the thermal performance of microchannels. 

3)Velocity vectors, velocity contours and temperature Contours needs to be explained and implemented well.

Need some minor corrections

Author Response

Response to Reviewer 2 Comments

This manuscript indicate that the wavy structure and fin shape have a significance effect on the heat sink performance. Heat transfer augmentation is observed in the wavy channels, especially coupled with different shaped fins. However, large penalty of pressure drops is also found in these channels. The diamond shaped fins yield the best heat transfer augmentation but the worst pumping performance, followed by the square, and circular shaped fins.

Overall, I recommend the manuscript be accepted after minor corrections,

Point 1: Pressure drop is an important issue in designing passages of coolant flows to repel high heat flux, but this issue is not discussed clear enough.

Response 1: Yes, the pressure drop is influenced by the blocking effect of the different shaped fins. The following statements are added in the revised manuscript:

“The circular shaped fins have the minimal blocking effect on the fluid because of its streamlined structures. However, the diamond shaped fins greatly change the direction of the main stream fluid, which have the maximum blocking effect on the fluid.” (Line 271-274)

Point 2: Explore the possibility and effect of geometric dimensions on the thermal performance of microchannels.

Response 2: Thank you for your kind suggestions. The geometric dimensions do have an important influence on the thermal performance of microchannels, and some studies which related to the amplitude and wavelength of wavy microchannels, the aspect ratio of wavy microchannels can be found in the literature. We have to say that the aim of this paper is to investigate the effect of the combination of wavy microchannel and different shaped fins on the heat transfer and fluid flow characteristics. Special attentions are paid to the comparison between the proposed microchannel heat sinks with the smooth straight channel. Therefore, the effect of geometric dimensions on the thermal performance of microchannels is beyond the scope of the current paper, which might be studied in the future work.

Point 3: Velocity vectors, velocity contours and temperature Contours needs to be explained and implemented well.

Response 3: The following statements are added in the revised manuscript:

“… the fluid in the middle of the microchannel has a higher velocity, while the fluid near sidewalls has a lower velocity.” (Line 244-246)

“…the reason can be explained by the enlarged heat transfer area for the wavy microchannels and different fins. Moreover, …” (Line 302-304)

Thank you very much for your comments and suggestions!

Prepared by Tingfang Yu, Xing Guo, Yicun Tang, Xuan Zhang, Lizhi Wang, and Tao Wu

June 30, 2023
